# Spaceflight and Ground-Based Microgravity Simulation Impact on Cognition and Brain Plasticity

**DOI:** 10.3390/ijms26199521

**Published:** 2025-09-29

**Authors:** Jiaqi Hao, Jun Chang, Yulin Deng

**Affiliations:** School of Medical Technology, Beijing Institute of Technology, Beijing 100811, China; jaq0714@gmail.com (J.H.); optics_chang@126.com (J.C.)

**Keywords:** microgravity, cognitive function, neural plasticity, synaptic plasticity, spaceflight analog models, neuroimaging

## Abstract

Microgravity exposure during spaceflight has been linked to cognitive impairments, including deficits in attention, executive function, and spatial memory. Both space missions and ground-based analogs—such as head-down bed rest, dry immersion, and hindlimb unloading—consistently demonstrate that altered gravity disrupts brain structure and neural plasticity. Neuroimaging data reveal significant changes in brain morphology, functional connectivity, and cerebrospinal fluid dynamics. At the cellular level, simulated microgravity impairs synaptic plasticity, alters dendritic spine architecture, and compromises neurotransmitter release. These changes are accompanied by dysregulation of neuroendocrine signaling, decreased expression of neurotrophic factors, and activation of oxidative stress and neuroinflammatory pathways. Molecular and omics-level analyses further point to mitochondrial dysfunction and disruptions in key signaling cascades governing synaptic integrity, energy metabolism, and neuronal survival. Despite these advances, discrepancies across studies—due to differences in models, durations, and endpoints—limit mechanistic clarity and translational relevance. Human data remain scarce, emphasizing the need for standardized, longitudinal, and multimodal investigations. This review provides an integrated synthesis of current evidence on the cognitive and neurobiological effects of microgravity, spanning behavioral, structural, cellular, and molecular domains. By identifying consistent patterns and unresolved questions, we highlight critical targets for future research and the development of effective neuroprotective strategies for long-duration space missions.

## 1. Introduction

In recent years, the continued advancement of manned space missions, the significant extension of spaceflight durations, and the transition of commercial space travel from concept to reality have increasingly exposed humans to the extreme physiological and psychological challenges posed by the space environment [1,2,3,4,5]. Among these, cognitive function—essential for executing complex tasks, ensuring flight safety, and accomplishing mission objectives—has emerged as a critical area of concern due to its susceptibility to alterations induced by microgravity. Accumulating evidence indicates that exposure to microgravity can result in a broad spectrum of cognitive impairments, including deficits in attention, executive function, and spatial memory [6,7,8,9]. These changes not only compromise astronauts’ performance during space missions but also present formidable challenges to health maintenance in long-duration exploration and future commercial spaceflight endeavors.

Extensive investigations have been conducted to explore the impact of microgravity on cognition through International Space Station (ISS) missions, animal spaceflight experiments, and ground-based analog models such as hindlimb unloading and head-down bed rest. These studies have collectively revealed the multifaceted manifestations of cognitive decline and have begun to uncover the underlying physiological mechanisms [10,11,12,13,14]. In particular, neuroplasticity—the neural basis of cognitive processes—has garnered increasing attention for its vulnerability and adaptive responses under microgravity conditions. Structural remodeling of brain tissue [15,16], alterations in synaptic morphology and function [17,18], and dynamic changes in neurotransmitters and neurotrophic factors offer valuable molecular and cellular perspectives on cognitive impairments observed in space [19,20]. Moreover, recent advances in proteomics and gene expression profiling have shed light on the intricate molecular cascades triggered by microgravity, including oxidative stress, mitochondrial dysfunction, and dysregulation of key signaling pathways [21,22]. These findings provide a theoretical foundation for the development of targeted prevention and intervention strategies aimed at mitigating spaceflight-associated cognitive deficits (Figure 1).

Despite the wealth of existing data, discrepancies arising from variations in experimental designs, model systems, and assessment methodologies continue to obscure a unified understanding of how microgravity affects cognitive function and neuroplasticity. Therefore, a comprehensive synthesis of current multidisciplinary evidence is urgently needed to clarify these mechanistic pathways. Such a review would be of substantial scientific value, offering critical insights into the neurobiological basis of cognitive vulnerability in space and guiding the design of effective countermeasures. In this context, systematically reviewing and analyzing the cognitive, neuroplastic, and molecular effects of microgravity environments will contribute to the construction of an integrated theoretical framework and propel future research in space neuroscience and aerospace medicine.

## 2. Impact of the Microgravity Environment on Cognitive Function

Previous studies have demonstrated that prolonged spaceflight exposure has been associated with deficits in attention, impaired executive function, and spatial memory deterioration [6,7,8,9]. These changes are believed to be partially attributable to microgravity-induced alterations in brain fluid distribution and disrupted mechanical signaling [1,2,3]. However, additional spaceflight-related factors, such as radiation, elevated noise levels, and isolation stress, also contribute to these cognitive disturbances [3,4,5]. To delineate the specific role of microgravity, findings from both spaceflight missions and ground-based analog models are reviewed below.

### 2.1. Evidence from Spaceflight Missions

Post-mission assessments of astronauts have consistently revealed perceptual and cognitive changes induced by altered sensory integration strategies under microgravity [23,24]. Specifically, the dominance of visual input over vestibular signals in weightlessness leads to a reorganization of sensory processing. This often manifests as spatial disorientation, altered depth and distance perception, impaired mental rotation ability, and in some cases, the so-called “time compression syndrome,” characterized by a subjective acceleration of time perception [25,26]. In a 2023 study, Salazar et al. reported spatial cognitive impairments in astronauts, including decreased performance in 3D navigation and object manipulation tasks, as well as impaired spatial working memory (SWM) [27]. These deficits were accompanied by delayed re-adaptation of posture control and balance following return to Earth. Hupfeld et al., in a 2021 study, further suggested that spaceflight induces adaptive neural effects, which may manifest either as cognitive deficits or compensatory plasticity [12]. Some findings support this view, reporting vestibular reweighting and postural instability, while certain cognitive functions—such as basic working memory—remained stable or even improved during missions, possibly due to neural compensation mechanisms [23,28,29]. Moore et al. conducted a longitudinal study on eight astronauts aboard the International Space Station (ISS) over a six-month mission. Their findings showed significant impairments in manual dexterity, dual-tasking, and motion perception. Compared with Earth-based controls, the astronauts exhibited notable declines in spacecraft operation performance [30]. According to reports from the National Aeronautics and Space Administration (NASA), nearly all astronauts returning from space missions report cognitive or psychomotor issues linked to central nervous system (CNS) alterations [10]. A landmark study—the NASA Twins Study—offered further compelling evidence regarding the cognitive risks associated with prolonged spaceflight. One key finding was that extended mission duration (12 months) resulted in cognitive impairments that persisted for up to six months post-flight, prompting NASA to categorize cognitive decline as a high-risk factor for long-duration missions [11].

Animal studies conducted in space have corroborated these findings, suggesting that the spaceflight environment—which includes microgravity along with other stressors—poses significant cognitive risks. Rodent models have shown impaired spatial memory, including increased error rates in object location recognition tasks and reduced performance in hippocampus-dependent tasks [22,31,32,33,34,35]. These cognitive changes have been directly associated with hippocampal structural and functional alterations observed in space-exposed animals. In addition, Ronca et al. reported stereotyped circling behavior in mice onboard the ISS, and in the same year, Kiffer et al. observed impaired spatial learning—evidenced by decreased performance in the Morris water maze [13,36].

While these findings cannot be solely attributed to microgravity as an isolated stressor, they underscore the importance of studying how the space environment, including microgravity, affects neuroplasticity and cognitive function in spaceflight contexts.

### 2.2. Complementary Evidence from Ground-Based Analog Studies

In addition to the unique experimental conditions provided by the International Space Station (ISS), several terrestrial analogs—such as parabolic flight, dry immersion, and head-down bed rest (HDBR)—have been widely employed to simulate the physiological and psychological effects of microgravity [35,37,38,39,40]. For example, Stahn et al. demonstrated that long-term social isolation combined with HDBR leads to volumetric reduction in the hippocampal dentate gyrus and impaired spatial memory performance [34]. Consistent findings suggest that simulated microgravity increases error rates and prolongs reaction times during visuospatial tasks [41,42]. There is also growing concern about the emotional and cognitive impacts of simulated microgravity, with some studies reporting a reduction in positive affect, heightened anxiety or fear, and task performance declines under HDBR conditions [43]. In contrast, Chen et al. observed that although short-term memory was impaired following a 45-day −6° HDBR intervention, depressive and anxiety-related symptoms remained largely unaffected [14]. Tomilovskaya et al., in a dry immersion study, reported an increase in errors on spatial orientation tasks, even during short-term exposures [44]. Similarly, parabolic flight experiments, which produce transient microgravity episodes, have shown temporary impairments in spatial updating abilities—likely linked to dysfunction in hippocampal–vestibular integration [44]. Cerebellar studies demonstrate that vestibular-sensitive Purkinje cells also encode neck proprioceptive cues, maintaining robust self-motion perception under altered gravity [45]. Similarly, Guillaud et al. (2020) reported that during parabolic-flight microgravity, vestibulo-spinal EMG responses were suppressed in the lower limbs but remained present in axial muscles despite the loss of otolithic input [46]. At the neural level, parabolic-flight studies have elucidated gravity’s effect on sensory coding. For example, Lecoq et al. (Aider’s group) cultured hippocampal neurons on a microfluidic chip and directly monitored calcium activity during parabolic flights [47]. They observed that sudden transitions between 1 g, 0 g, and 2 g phases produced rapid, gravity-dependent changes in neuronal calcium signaling [47]. In humans, psychophysical work shows gravity-dependent perceptual biases: upward isometric forces feel larger than identical downward forces, an illusion amplified in hypergravity and eliminated in microgravity [48]. These findings, together with single-unit recordings in vestibular pathways, indicate that gravity provides a modality-independent cue that shapes the brain’s internal model of orientation and movement [48,49]. Indeed, Pompeiano and colleagues demonstrate that neural representations of self-motion rapidly adapt to altered gravito-inertial loads [50].

Rodent studies using ground-based microgravity analogs, particularly the hindlimb unloading (HU) model, have provided critical insights into the neural consequences of weightlessness. HU is a well-established method to simulate microgravity and consistently induces deficits in spatial learning and memory, as evidenced by impaired performance in the Morris water maze (MWM) and Barnes maze tasks, including prolonged escape latencies and reduced navigation efficiency [31,35,51,52,53,54,55]. These cognitive impairments are closely linked to hippocampal dysfunction, encompassing synaptic plasticity deficits, impaired cholinergic signaling, and reduced adult neurogenesis. Importantly, accumulating evidence suggests that these effects are not simply attributable to generalized stress responses. For example, Hernandez et al. reported that 7-day HU altered regional norepinephrine levels in the inferior olive, cerebellum, and substantia nigra without elevating plasma corticosterone [56], indicating specific neuromodulatory changes rather than systemic stress activation. Similarly, HU-induced reductions in hippocampal progenitor cells have been demonstrated by Groussard et al. and Yasuhara et al. [57], whereas interventions such as exercise or plantar stimulation effectively prevented or reversed these neurogenic deficits [58,59]. Together, these findings underscore that gravity deprivation per se disrupts hippocampal plasticity and monoaminergic regulation, thereby impairing spatial cognition through mechanisms distinct from classic stress pathways.

### 2.3. Controversies and Future Directions

Despite widespread recognition of microgravity-induced cognitive impairments, some inconsistencies remain. For instance, Temple et al. reported in 2002 that exposure to microgravity had minimal long-term effects on spatial learning and memory in developing rats (postnatal day 8 or 14), with most cognitive changes resolving rapidly after re-exposure to Earth’s gravity [60]. Wollseiffen et al. further suggested that the effects of spaceflight on cognition are multifactorial, and even observed transient enhancements in certain cognitive abilities under short-term microgravity conditions [61].

These discrepancies may stem from variations in study design, including differences in the duration of exposure, the choice of ground-based model, species, or developmental stages of experimental animals, and the specific cognitive assessments employed. As a result, the comparability of findings across studies remains limited. Future investigations should adopt more standardized, integrated approaches to systematically evaluate the cognitive and neurobiological consequences of microgravity. Establishing such methodological consistency will be critical to resolving current controversies and developing effective countermeasures for cognitive protection during spaceflight.

Figure 2 summarizes the timeline of microgravity-induced neural plasticity, highlighting effects that emerge acutely (minutes–hours), in the short term (days), and over longer durations (days–weeks), as well as the post-flight readaptation phase.

## 3. Current Understanding of Microgravity-Induced Effects on Neuroplasticity

The investigation of how spaceflight impacts the nervous system has both scientific and practical implications. Neuroplasticity forms the foundation for the wide array of physical, psychological, and behavioral changes that occur within the human nervous system during space missions [70]. It encompasses changes at both the systems level—such as alterations in large-scale brain networks—and the cellular level, including structural and functional synaptic plasticity [71,72]. Neuroplasticity reflects the brain’s and individual neurons’ capacity to adapt to novel stimuli or environmental changes and is fundamental to learning and memory processes [65,72]. Understanding these adaptive mechanisms may yield valuable insights into the neurobiological underpinnings of cognitive alterations observed during spaceflight. This section reviews current evidence regarding the impact of spaceflight and ground-based microgravity simulations on neuroplasticity, with a focus on changes in brain structure, synaptic plasticity, and molecular signaling mechanisms.

### 3.1. Structural Changes in Brain Tissue

Spaceflight-induced alterations in brain structure have been frequently observed in astronauts. The effects of microgravity on brain anatomy and function are often assessed using magnetic resonance imaging (MRI), a powerful neuroimaging technique with high spatial resolution. MRI provides detailed structural, functional, and biochemical information from specific brain regions and is widely used to monitor dynamic neuroplastic responses. Functional MRI (fMRI), which measures brain activity and connectivity based on blood oxygenation level-dependent (BOLD) signals, is particularly valuable for mapping region-specific neural activation with sub-centimeter spatial resolution [73].

Over the past decade, numerous pre- and post-flight MRI studies have revealed widespread alterations in brain structure and connectivity attributable to microgravity exposure [16,18,66,74]. In a landmark 2016 study, Koppelmans et al. assessed MRI scans of 27 astronauts and reported significant structural brain changes following spaceflight. These included widespread gray matter volume reductions, focal increases in gray matter in sensorimotor regions, and white matter disruptions in areas critical for visuomotor coordination and high-order visuospatial processing [75]. Lee et al. further demonstrated reduced integrity of key white matter tracts, including the superior and inferior longitudinal fasciculi [76]. However, subsequent studies have presented more nuanced findings. For example, Jillings et al. observed increased cerebellar white matter volume immediately after spaceflight, with continued increases up to seven months post-return, while gray matter exhibited morphometric changes without clear evidence of tissue loss [77]. Consistent findings by Van Ombergen and colleagues using both spaceflight data and ground-based microgravity simulations revealed brain gray matter volume shifts, cerebrospinal fluid (CSF) redistribution toward the head, and reduced resting-state functional connectivity in vestibular-related cortical areas [64,65,68,69]. These alterations were associated with diminished motor, learning, and memory performance. It has been hypothesized that impaired vestibular input contributes to spatial memory deficits [78]. Roberts et al. compared pre- and post-mission brain scans from 18 astronauts and reported frequent observations of central sulcus narrowing, upward brain displacement, and CSF space reduction at the brain vertex [79]. Lateral ventricular enlargement was also commonly reported, with volume increases ranging from 7% to 25%, and recovery often requiring months to years [17,68,80,81,82]. Further structural changes reported by Liao et al. and Arone et al. included thinning of the right occipital and bilateral temporal cortices, thalamic volume reductions, and pituitary gland deformation [83]. Riascos et al. compared short-duration (one month) and long-duration (six months or more) space missions and found no significant relationship between flight duration and the extent of brain changes [84]. Conversely, another study found that longer missions aboard the ISS were associated with more extensive fluid shifts [82].

These findings collectively suggest that spaceflight exposure may alter brain tissue structure, CSF distribution, and functional connectivity to varying degrees, with microgravity likely playing an important but not exclusive role. However, due to the limited availability of in-flight data, small sample sizes, and methodological inconsistencies across studies, it remains difficult to fully characterize these effects. Ground-based microgravity simulations provide a valuable complementary platform for further elucidating these changes.

Ground-based studies using head-down bed rest (HDBR) and MRI have demonstrated similar findings. Roberts et al. reported that in eight long-term HDBR participants, the brain could shift within the cranial cavity in response to gravity changes, and such morphological displacements may underlie associated changes in brain function [15]. Liao et al. and Zhou et al. used resting-state fMRI to evaluate 16 healthy male volunteers following 45 days of HDBR. They found that the salience network—particularly involving the anterior insula (aINS) and the midcingulate cortex (MCC)—exhibited disrupted connectivity, likely affecting cognitive and emotional information processing. These alterations may contribute to cognitive deficits observed during spaceflight [66,67]. Cassady et al. conducted a longitudinal study using fMRI at seven time points during and after HDBR and reported significant connectivity changes in vestibular, somatosensory, and motor networks, which were correlated with changes in sensorimotor performance and spatial working memory. Interestingly, these neural changes were not always detrimental and were sometimes suggestive of adaptive reorganization [85]. Li et al. observed that after 30 days of HDBR, participants exhibited significant gray matter volume reductions in the bilateral frontal lobes, temporal poles, parahippocampal gyri, insulae, and right hippocampus. In contrast, increased gray matter volume was noted in the cerebellar vermis, bilateral paracentral lobules, right precuneus, and primary sensorimotor cortices. Changes in fractional anisotropy (FA) were also observed across multiple white matter tracts [86].

In summary, both spaceflight and ground-based studies consistently indicate that microgravity induces dynamic changes in brain structure, CSF distribution, and functional connectivity. These structural and functional brain adaptations likely underlie cognitive alterations observed during and after spaceflight. MRI-based findings provide a theoretical framework for developing countermeasures to protect and rehabilitate astronauts’ cognitive and neurological health upon return to Earth.

### 3.2. Synaptic Plasticity Alterations

In addition to inducing macrostructural changes in brain anatomy, microgravity exposure can also lead to neuronal alterations at the cellular level. Although current studies on synaptic plasticity under microgravity are largely limited to spaceflight and ground-based rodent models, and cross-species extrapolation remains challenging, these findings offer critical insights into the potential mechanisms by which neurons adapt—or fail to adapt—to altered gravity.

#### 3.2.1. Structural Synaptic Plasticity

At the cellular level, microgravity affects neuronal structure through multiple physical cues, including altered hemodynamics, hydrostatic pressure, shear stress, tissue tension, and impaired mass transfer and membrane permeability. These mechanical shifts influence both cellular morphology and cytoskeletal dynamics [87].

Studies in flight animals have demonstrated that prolonged exposure to the spaceflight environment, including weightlessness, can be associated with either degenerative or compensatory structural changes in brain regions responsible for sensory processing, motor control, and cognition—including the hippocampus, parietal cortex, and cerebellum. During the Neurolab mission, experiments in rats and fish showed marked alterations in synaptic morphology within the hippocampus, cerebral cortex, and cerebellum, typically manifesting as a decrease in synapse number and a pronounced sparsification of dendritic arbors; consequently, the overall complexity of the local synaptic networks was reduced, whereas synaptic connectivity in vestibular nuclei increased significantly [88,89,90,91]. This pattern has been interpreted as a compensatory response to the loss of gravitational input [91]. In support of this notion, Belichenko and Krasnov reported a variable yet significant increase in dendritic spine density on layer V pyramidal neurons of the sensorimotor cortex in tail-suspended rats subjected to a 14-day spaceflight aboard “Kosmos-2044” [91]. By contrast, data from NASA’s Spacelab-2 revealed that microgravity provoked axonal terminal degeneration and an abnormal distribution of axonal endings within the somatosensory cortex, as documented by D’iachkova [92].

Experimental models have also provided direct evidence supporting these findings. Yasuhara et al. reported that a 2-week hindlimb suspension (HS) model significantly reduced neurogenesis in the subventricular zone and dentate gyrus of rats. Notably, neither physical exercise nor recovery reversed the HS-induced decline in neurogenesis or the downregulation of neurotrophic factors [57]. Similarly, Xiang et al. observed a marked decrease in dendritic spine density within both the dentate gyrus (DG) and CA1 region of the hippocampus in mice subjected to long-term hindlimb unloading [20]. Ranjan et al. showed that rats exposed to 14 days of simulated microgravity (SMg) exhibited significant decreases in the average area, perimeter, synaptic cleft width, and active zone length of CA1 hippocampal neurons. Conversely, dendritic arbor complexity and spine number were significantly increased, suggesting a compensatory remodeling process induced by microgravity exposure [93]. By contrast, exposures to hypergravity (e.g., centrifugation or 2–3 g) elicit different neural responses. Brungs et al. demonstrated that exposure of hippocampal neurons to 2 g hypergravity for 24 h resulted in the formation of more mature synaptic connections, with a 30% increase in neurite number and a 20% increase in neu-rite length distribution [94]. Human parabolic-flight work shows that reaching movements remain accurate in 0g but become systematic in 1.8 g participants underreach targets due to incomplete reorganization of muscle commands [95]. Opsomer et al. found that the “gravity illusion” (upward force feel) is enhanced in 2 g [48]. In rodents, 2 g for 24 h actually enhances hippocampal synaptic growth (more neurites) [95]. Overall, these studies illustrate how organisms respond and adapt to altered gravity environments, with hypergravity research in particular providing valuable insights into how biological systems adjust to increased gravitational load.

Collectively, these findings indicate that microgravity-induced structural changes at the synaptic level may disrupt the integrity of neural circuits and compromise functional connectivity, thereby contributing to cognitive dysfunction (Figure 3).

#### 3.2.2. Functional Synaptic Plasticity

Microgravity also alters functional aspects of synaptic plasticity, including neurochemical signaling and electrophysiological responses.

Animal flight experiments have shown that prolonged spaceflight, where weightlessness is a key component among multiple stressors, markedly disrupts the homeostatic regulation of central neurotransmitters [54,96,97,98,99,100]. Both the month-long Russian Bion-M1 mission and the 18.5-day Biosatellite Cosmos-1129 mission revealed that extended space-flight profoundly alters the genetic control of brain dopamine (DA) while exerting only a moderate effect on 5-hydroxytryptamine (5-HT) levels [101,102]. In addition, Zhu and Desiderio reported a distinctive neuropeptidergic stress response after just five days of acute space-flight: concentrations of methionine-enkephalin (ME) and substance P (SP) in the rat neurohypophysis declined, whereas β-endorphin (BE) remained unchanged [103]. Using differing hind-limb unloading (HU) exposures in mice, Zhang et al. demonstrated that learning-and-memory deficits arising after more than 14 days of HU involve both cholinergic dysfunction and oxidative damage, and that the severity of impairment increases with HU duration [54]. Proteomic analyses further indicate that differential proteins after HU are enriched in pathways related to synaptic transmission and the regulation of glutamate (Glu) and γ-aminobutyric acid (GABA), supporting the view that hippocampal excitotoxicity under long-term microgravity underlies cognitive deficits [19,104,105].

Neurotrophic factors are likewise critical for neuronal survival and plasticity. Brain-derived neurotrophic factor (BDNF) is widely expressed in the brain and plays key roles in neuronal growth and synaptogenesis [106,107]. Nonetheless, studies in flight animals have reported no significant change in BDNF levels [108,109]. To date, only ground-based work has shown down-regulation of CREB/BDNF-related proteins after 28 days of HU in mice [20]. In contrast, long-duration space flight exerts pronounced effects on glial cell line-derived neurotrophic factor (GDNF) and cerebral dopamine neurotrophic factor (CDNF). Tsybko and colleagues found that one month in microgravity decreased GDNF gene expression in the striatum and hypothalamus but increased it in the frontal cortex and raphe nuclei, while simultaneously lowering CDNF expression in the substantia nigra yet elevating it in the raphe nuclei [96]. Moreover, Santucci et al. reported a trend toward reduced nerve growth factor (NGF) expression in the cerebellum, hippocampus, cortex, and adrenal glands of mice exposed to three months of simulated microgravity [96].

Endocrine alterations accompany these neurochemical changes. Work by Angeloni and Demonstis showed that long-duration space-flight heightens hypothalamic–pituitary–adrenal (HPA) axis activity in astronauts, a response suggestive of chronic stress [110]. Likewise, Maurice et al. demonstrated that simulated weightlessness significantly lowers circulating hormone indices, including plasma renin activity and atrial natriuretic peptide [111].

Finally, long-term potentiation (LTP)—the electrophysiological basis of hippocampus-dependent learning and memory [112]—appears particularly vulnerable to altered gravity. Across a spectrum of gravitational conditions, animal studies consistently report concurrent LTP impairment and behavioral deficits [97,113,114], including hind-limb unloading and other microgravity models [20,115,116]. Collectively, these findings underscore the heightened sensitivity of the hippocampus to gravitational change (Figure 3) [93].

#### 3.2.3. Molecular Mechanisms

Proteomic and transcriptomic studies have further revealed that the molecular mechanisms underlying HU-induced impairments in synaptic plasticity are likely closely associated with pathological processes such as enhanced oxidative stress and the suppression of synaptic function-related proteins [109].

Bioinformatic analyses combined with experimental validation by Wu et al. demonstrated that during simulated long-term spaceflight environment (SLSE)-induced cognitive impairment, the PI3K-Akt-mTOR signaling pathway is significantly inhibited, whereas the MAPK pathway is activated [117,118]. Supporting this, Wang et al. confirmed that simulated microgravity (SM) significantly impacts the expression of hippocampal metabolic proteins, highlighting its profound effects on brain biochemistry [21]. Similar alterations in MAPK signaling have been identified in rat models of microgravity, where its interaction with the SNARE complex is thought to be a key factor affecting learning and memory functions [119]. Frigeri et al. reported that hindlimb unloading causes significant changes in brain gene expression, particularly in genes associated with learning, memory, and synaptic transmission. Among the upregulated genes, the largest proportion belonged to the TIC category (transport of small molecules and ions into cells), whereas downregulated genes were predominantly classified under JAE (cell junctions, adhesion, and extracellular matrix organization) [119]. These effects appear to be especially pronounced in the hippocampus, a brain region critically involved in learning, memory, and spatial navigation. Numerous studies have reported that microgravity exposure is associated with altered expression of genes in the hippocampus related to both cognitive function and antioxidant defense [20,120]. Zhai et al. proposed that the downregulation of postsynaptic function-related proteins and BDNF/TrkB signaling is a central mechanism underlying learning and memory impairments under long-term microgravity exposure. They further suggested that high-frequency repetitive transcranial magnetic stimulation (rTMS) may mitigate such damage [116].

Due to the lack of direct molecular biomarker data from human astronauts, further research is required to evaluate the translational potential of these animal-based findings.

## 4. Countermeasures and Future Research Directions

### 4.1. Countermeasures

Developing effective countermeasures is paramount. Exercise is the cornerstone intervention. In-flight resistive and aerobic exercise maintain cardiovascular fitness and have neuroprotective effects: in rodents, running reverses HU-induced reductions in hippocampal neurogenesis and supports synaptic integrity [58]. Artificial gravity (AG) is a promising strategy to supply a normal gravitational load. A recent 60-day HDBR study found that daily centrifugation (1 g at the center) altered brain connectivity trajectories: control subjects showed pathologically increased parietal–somatosensory connectivity (likely maladaptive), whereas the AG group normalized this connectivity and exhibited less balance decline post-HDBR [121]. This suggests AG may counteract the sensory deprivation of microgravity by reweighting somatosensory input. Pharmacology is a less-explored but interesting area: agents that modulate GABA or glutamate, or support neurotrophic signaling, could theoretically ameliorate microgravity-induced neurotransmitter imbalances. Finally, combined interventions (integrating exercise, intermittent AG, cognitive training, and targeted drugs) may be required to preserve neural health on deep-space missions.

### 4.2. Future Research Directions

#### 4.2.1. Expanding Sample Sizes and Conducting Longitudinal Studies

Current studies on the cognitive effects of microgravity are mostly based on small samples of astronauts or rodent models, limiting statistical power and increasing susceptibility to individual variability. Future research should leverage international collaboration to establish large-scale, multi-center aerospace medicine databases. Longitudinal studies tracking cognitive changes across pre-flight, in-flight, and post-flight stages are essential. Integration of behavioral assessments with electrophysiological recordings and neuroimaging techniques—such as functional MRI (fMRI), electroencephalography (EEG), and functional near-infrared spectroscopy (fNIRS)—may reveal causal links between cognitive impairment and structural or functional brain alterations.

#### 4.2.2. Standardization and Translational Application of Ground-Based Models

Ground-based microgravity analogs, including head-down bed rest (HDBR), dry immersion, parabolic flight, and hindlimb unloading (HU), are widely used to simulate microgravity effects. However, these models currently lack standardized exposure parameters, simulation depth, and assessment frameworks. Future efforts should aim to standardize experimental protocols to improve reproducibility and cross-study comparability. Moreover, establishing a robust correspondence between simulated conditions and real spaceflight data is critical for translational validity. The development of advanced platforms, such as precision-controlled simulators and 3D organoid-based systems, may further expand the depth of research and the scope of its applications.

#### 4.2.3. Dynamic Tracking of Synaptic Structural and Functional Plasticity

Neural plasticity is central to understanding microgravity-induced brain changes, yet most studies rely on static observations at isolated time points. Future work should utilize advanced imaging techniques—such as two-photon microscopy and super-resolution imaging—combined with neural circuit tracing tools, to enable real-time visualization of synaptic remodeling. Integration of single-cell RNA sequencing, spatial transcriptomics, and proteomics can help characterize subtype-specific neuronal responses to microgravity and uncover the spatiotemporal-functional relationships between synaptic plasticity and cognitive behavior.

#### 4.2.4. Investigating Reversibility and Intervention Strategies for Cognitive Impairment

The reversibility and recovery mechanisms of microgravity-induced cognitive dysfunction remain unclear. Longitudinal studies encompassing both intervention and recovery phases are needed to determine the extent to which synaptic plasticity and network functions can be restored. Multimodal countermeasures—including neuromodulation (e.g., transcranial magnetic stimulation, tDCS), behavioral training (e.g., attention or working memory exercises), and pharmacological approaches (e.g., neurotrophic factor enhancers, antioxidants)—should be evaluated for efficacy and mechanisms under simulated microgravity, thereby informing in-flight intervention strategies.

#### 4.2.5. Promoting Interdisciplinary Collaboration and Clinical Translation

Research on microgravity-induced cognitive deficits has translational value beyond space missions, offering novel models for understanding neurodegenerative diseases, cognitive decline, and aging on Earth. Future efforts should promote interdisciplinary collaboration across aerospace medicine, neuroscience, geriatrics, artificial intelligence, and bioinformatics. Integrating machine learning with multi-omics analysis will facilitate the identification of critical pathways and predictive biomarkers, advancing early detection and personalized intervention strategies for cognitive disorders.

## 5. Conclusions

Overall, the impact of microgravity on human cognitive function and neural plasticity has emerged as a critical topic in space medicine and neuroscience. Extensive evidence from both space missions and ground-based analog models indicates that microgravity exposure impairs multiple dimensions of cognition, including attention, executive function, and spatial learning and memory. These deficits are closely associated with structural remodeling of the brain, synaptic plasticity impairments, and a range of dysregulated molecular mechanisms. Neuroimaging studies have revealed alterations in brain volume and functional connectivity, while cellular and molecular analyses further highlight synaptic degeneration, neurotransmitter imbalance, enhanced oxidative stress and inflammatory responses, and mitochondrial dysfunction as potential contributors to cognitive decline. Additionally, proteomic and transcriptomic data emphasize the role of mitochondrial impairment and disrupted signaling cascades in the pathogenesis of microgravity-induced cognitive deficits.

Despite these advancements, current research is limited by small sample sizes, heterogeneous models, and inconsistent evaluation parameters. Human data remain scarce, and the reversibility of cognitive dysfunction as well as the effectiveness of potential interventions have yet to be systematically validated. Future efforts should focus on standardized, multi-center, and longitudinal studies, the development of refined ground-based simulation platforms, and integrated multi-omics approaches to elucidate the reversibility and molecular underpinnings of microgravity-induced cognitive impairment. These strategies will be essential for informing in-flight countermeasures and facilitating translational applications on Earth.

## Figures and Tables

**Figure 1 ijms-26-09521-f001:**
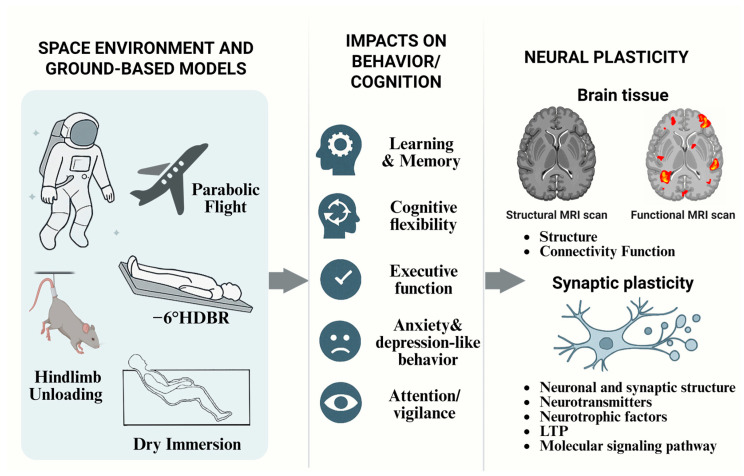
An overview of spaceflight missions on the International Space Station (ISS), animal spaceflight experiments, and ground-based analogs such as hindlimb unloading and head-down tilt bed rest, illustrating the effects of microgravity on cognition. This includes changes in structural and functional connectivity of the brain, as well as alterations in various aspects of synaptic plasticity. These findings provide a theoretical basis for developing targeted preventive and interventional strategies to mitigate spaceflight-associated cognitive impairments.

**Figure 2 ijms-26-09521-f002:**
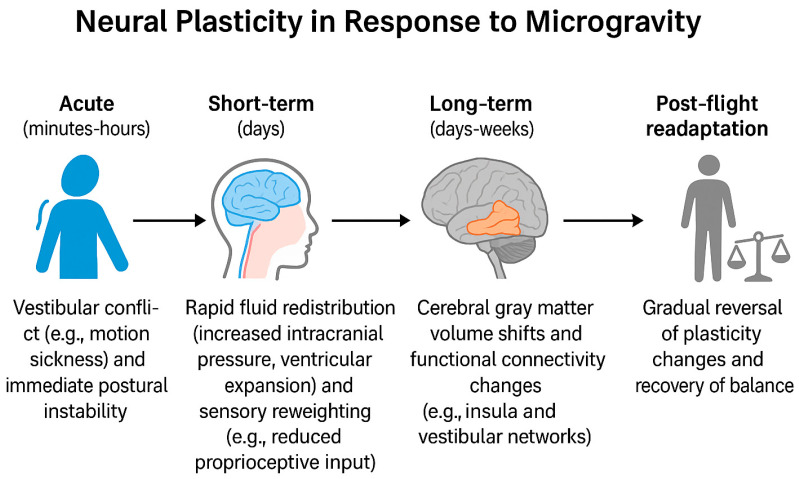
Neural adaptation timeline [11,23,24,25,26,47,59,62,63,64,65,66,67,68,69].

**Figure 3 ijms-26-09521-f003:**
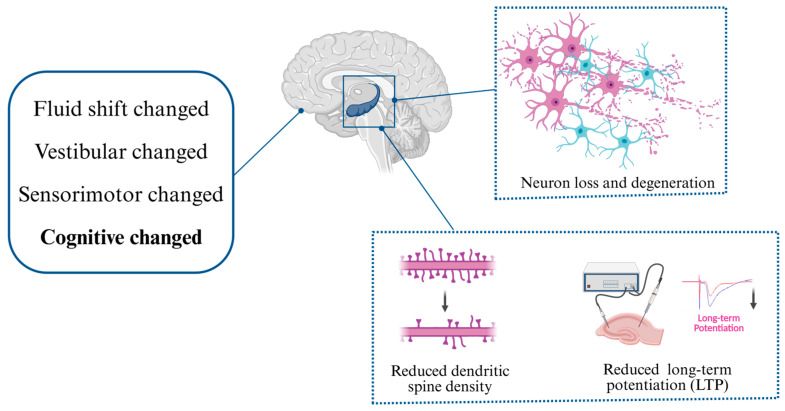
Summary of brain changes and hippocampal neuroplasticity alterations under microgravity exposure. Multiple studies have reported the loss and degeneration of hippocampal neurons, reduced dendritic spine density, and decreased long-term potentiation (LTP) after exposure to microgravity.

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
