# Peer review of "Spaceflight and Ground-Based Microgravity Simulation Impact on Cognition and Brain Plasticity"

_ijms, 2025, doi:10.3390/ijms26199521_

Round 1
Reviewer 1 Report
Comments and Suggestions for Authors
dear colleagues,
The manuscript proposal is important for several research communities, including space biology and neurosciences. While the recently published review (Wuyts et al, Nat Rev Neurosci 2025) offers an overview of the literature in the field, it was incomplete (due to the subject and format). Then, it is possible to improve your MS to make a more complete and comprehensive literature review.
Firstly, it is important to revisit the work on parabolic flight on aspects of proprioception in humans and the perception of gravitational changes by neurons during parabolic flight. You could cite the work of the groups as Cullen, Carriot, Tagliabue, Denise Guillaud and Aider. There are also bed-rest models (which are discussed in the review cited above). On the other hand, animal models have provided fundamental results on issues such as the differentiation between stress and the effects of microgravity, and in finding causal links at the cellular or molecular level (comparison between stress and hindlimb unloading or hypergravity exposure, studies on changes in monoamine distribution, alterations in neurogenesis, etc.). Many groups, particularly in Europe, have worked on these subjects. Studies conducted in hypergravity could be reported in order to explain how organisms respond to gravitational change.
It seems more appropriate to me to cite research articles rather than reviews of issues, or to do so in a specific paragraph, explaining the relevance of each.
A figure could be included showing the timeline of the biological response (of the nervous system) to space flight or exposure to simulated microgravity.
The effects of countermeasures are not discussed, and I believe that a paragraph could focus on this point in the perspectives, which are otherwise very constructive and interesting. My criticisms are intended to make the review as comprehensive and relevant as possible; the work already accomplished is remarkable.
It should also be very necessary to update the literature, in fact there is a problem with the formatting of the bibliography: many references appear with only the initials of the authors' names, which is not correct. There are typos, such as in lines 32 and 210.
The manuscript needs to be improved for publication.
Author Response
Response to Reviewer
We sincerely thank the reviewer for the thorough and constructive comments, as well as for the positive evaluation of our work. We have carefully revised the manuscript to address all the points raised. Our detailed responses are as follows:
Comment 1:
It is important to revisit the work on parabolic flight regarding proprioception in humans and neuronal perception of gravitational changes (Cullen, Carriot, Tagliabue, Denise Guillaud, Aider).
Response:
We appreciate this suggestion. We have revised the manuscript to include key studies on parabolic flight related to proprioception and neuronal responses, citing the recommended authors and groups (see Section 2.2, the new text is marked in red font).
Comment 2:
Bed-rest models should be included, as well as animal models that distinguish between stress and microgravity effects, and studies on monoamine distribution, neurogenesis, and hypergravity exposure.
Response:
We agree and have expanded the relevant sections to include bed-rest studies and additional animal model findings. We have also added references to work on stress vs. microgravity effects, monoamine alterations, neurogenesis, and hypergravity exposure (see Section 2.2 and 3.1.1, the new text is marked in red font).
Comment 3:
It seems more appropriate to cite research articles rather than reviews, or to place reviews in a separate section with explanation of their relevance.
Response:
Thank you for the reviewer’s valuable suggestion. We fully agree on the central role of primary research in supporting specific experimental conclusions; throughout the manuscript, primary research articles are preferentially cited where concrete experimental evidence or mechanistic detail is discussed. At the same time, we have also cited several high-quality reviews (for example, Wuyts 2025, Yin 2023, and Stahn 2023) because they offer concise, structured syntheses of the field, clarify methodological differences, and highlight gaps in knowledge—features that help readers quickly grasp the broader context and trace the relevant primary literature.
For readability and to serve different reader needs, we therefore adopted a balanced citation strategy in which primary studies underpin specific claims while reviews are used for broader synthesis and context. If the reviewer prefers that we supplement particular statements with additional primary references or add a short paragraph summarizing our criteria for citing reviews versus original studies, we would be glad to make those additions.
Thank you again for the constructive and helpful comment.
Comment 4:
A figure could be included showing the timeline of the biological response of the nervous system to spaceflight or simulated microgravity.
Response:
We appreciate this suggestion. We have added a new figure (Figure 2) summarizing the timeline of nervous system responses to spaceflight and ground-based microgravity models.
Comment 5:
The effects of countermeasures are not discussed; a paragraph on this should be added in the perspectives.
Response:
We agree with this valuable point. We have added a dedicated paragraph on countermeasures in the Section 4 (see Section 4.1).
Comment 6:
It is necessary to update the literature; also, the bibliography formatting should be corrected (authors’ names), and typos (e.g., in lines 32 and 210) should be fixed.
Response:
The reference formatting has been corrected to include full author names, and the noted typos (lines 32 and 210) as well as others identified during proofreading have been corrected.
We thank the reviewer once again for the insightful and constructive comments. These suggestions have significantly improved the quality, comprehensiveness, and clarity of our manuscript.
Reviewer 2 Report
Comments and Suggestions for Authors
The present review has the aim to sum-up the scientific knowledge on the effect of spaceflight or simulated microgravity on different aspects of brain morphology and neuronal functions. This topic is relevant having a large impact in the field of space exploration and space biomedicine. Overall, the manuscript is well written and well discussed. The figures are in line with the text and add clarity to the manuscript. For these reasons I consider that this manuscript deserves to be published, even if I have some general considerations that need to be addressed by the authors to improve the clarity of the manuscript.
Specific points:
-In several parts of the text the authors attribute to microgravity the brain and neuronal changes observed during space flight (for instance see the text in line 76, 113, 217, 271, 311). I do not consider correct this direct attribution of causality, since during spaceflight other hazards (such as space radiation, noise, cyrcadian rythm alteration, stress work condition...) can contribute to the brain and nervous system alterations (as stated even by the authors is some sentences of the manuscript). I suggest to rephase the sentences indicated to avoid misleading statemets
- In line to what already said even the title seems to me that need to be rephrased, since attribute only to microgravity the brain alterations described in the review. I suggest to change the title as follow: "Spaceflight and ground-based microgravity simulation impact on cognition and brain plasticity" or using another sentence in which is clear that the reported literature is not only on microgravity simulation but also on space mission in which the complex space exposome includes several hazards (microgravity is just one of them) that act together impacting on living systems.
< !--EndFragment -->
Author Response
We sincerely thank the reviewer for the positive evaluation of our work and for the constructive suggestions. Our point-by-point responses are provided below:
Comment 1:
In several parts of the text the authors attribute to microgravity the brain and neuronal changes observed during space flight … I suggest to rephrase the sentences indicated to avoid misleading statements.
Response:
We agree with the reviewer that it is not correct to attribute these changes solely to microgravity. We have revised the indicated sentences (lines 76, 113, 217, 271, and 311) to state that the effects are associated with the spaceflight environment, where microgravity is an important but not exclusive factor.
Comment 2:
Even the title seems to need rephrasing, since it attributes only to microgravity the brain alterations described in the review. I suggest changing the title to make clear that the reported literature includes both microgravity simulations and space missions.
Response:
We appreciate this valuable suggestion. We have revised the title to:
“Spaceflight and ground-based microgravity simulation impact on cognition and brain plasticity.”
This better reflects the scope of the review and acknowledges that spaceflight includes multiple concurrent hazards in addition to microgravity.
Reviewer 3 Report
Comments and Suggestions for Authors
I am happy to recommend this review of “Microgravity and the brain” for publication in IJMS. It well fulfils its promise in the Abstract both to be a synthesis of the effects of microgravity on cognition and on the brain’s neurobiology, and to highlight critical targets for future research and effective neuroprotective strategies for humans in space. This review delivers strong recommendations for coordination and standardization in studies that lead to in-flight countermeasures against the deconditioning produced in human subjects by microgravity. That these recommendations are taken seriously is vital for an off-world era in commercial space projects that is closer upon us than is widely understood.
This relatively brief review has been well researched and abundantly cites relevant studies which have a broad international base. It is composed clearly and makes the outcome of studies, even when these may conflict, clear and concise. I detect skilled mentoring in this review’s conception and production.
For these reasons I find this work to be “a significant contribution to the field”. At issue is not only the efficiency of humans in space but their safe responses in emergencies. I would urge the authors and the institute they represent to make their recommendations for directing research clear to the government and commercial spaceflight concerns that will soon enough be sending general workforces into space.
Author Response
Dear Reviewer,
We sincerely thank you for your positive and encouraging comments on our manuscript. We are very pleased that you find our review a significant contribution to the field and appreciate your recognition of our recommendations for future research and countermeasures in spaceflight. Your supportive feedback is highly motivating for our work.
Sincerely,
Jiaqi Hao
On behalf of all authors
Round 2
Reviewer 1 Report
Comments and Suggestions for Authors
Dear authors,
Thank for the addition of several paragraphs to be more complete.
I see one type line 149.
Author Response
We sincerely thank the reviewer for pointing this out. We have carefully checked line 149 and revised the sentence to ensure clarity and correctness.